# Concomitant Autoimmune Liver Disease and Hepatic Actinomycosis: A Diagnostic Challenge—Brief Report and Review of the Literature

**DOI:** 10.3390/ijms26199420

**Published:** 2025-09-26

**Authors:** Giulia Manni, Martina Pambianco, Chiara Sicuro, Erica Franceschini, Alessandra Pivetti, Laura Bertoni, Veronica Bernabucci, Marcello Bianchini, Barbara Lei, Federico Ravaioli, Antonio Colecchia

**Affiliations:** 1Gastroenterology Unit, University of Modena & Reggio Emilia, 41124 Modena, Italy; giuliamannimed@gmail.com (G.M.);; 2Infectious Disease Clinic, AOU Modena, 41124 Modena, Italy; 3Gastroenterology Unit, AOU Modena, 41124 Modena, Italy; 4Department of Surgery, Medicine Dentistry and Morphological Sciences with Interest in Transplant, 41124 Modena, Italy; 5Department of Medical and Surgical Sciences, University of Bologna, 40138 Bologna, Italy

**Keywords:** actinomycosis, IgG4-related autoimmune liver disease, liver mass, *Actinomyces* species, molecular techniques

## Abstract

Hepatic actinomycosis (HA) and IgG4-related inflammatory pseudotumors are rare and often overlooked causes of liver mass, which can easily be misdiagnosed as primary liver cancer or metastasis. Diagnosis is arduous due to unspecified clinical and radiological features and the fact that histology is not always conclusive. In cases of actinomycosis, the use of molecular diagnostic techniques—such as polymerase chain reaction (PCR) for bacterial DNA—can aid in establishing a definitive diagnosis, especially when conventional cultures are non-diagnostic. We present a case report of one of our patients who was incidentally diagnosed with a hepatic lesion presenting aspecific radiological features. Since radiological imaging was inconclusive, a biopsy was performed, and a diagnosis of IgG4 related hepatic inflammatory pseudotumor was then made. Because of the disease progression, during immunosuppressive therapy, our diagnosis was questioned and a new liver biopsy was carried out. At the end, it took three consequent biopsies to finally find out the presence of an *actinomyces* infection.

## 1. Introduction

Actinomycosis is a rare, chronic, and slowly progressive infection caused by anaerobic Gram-positive bacilli of the genus *Actinomyces*. These bacteria are part of the normal flora of the oropharynx, gastrointestinal, and genitourinary tracts, making humans the natural reservoir for pathogenic *Actinomyces* species [1]. Infection usually occurs following a disruption of the mucosal barrier, allowing the bacteria to invade adjacent tissues. Once established, they form microscopic or macroscopic aggregates of tangled filaments surrounded by neutrophilic infiltrates, which may develop into visible yellowish structures known as sulfur granules [2,3]. However, classic histological and microbiological diagnosis is often challenging; therefore, molecular assessments such as next-generation sequencing (NGS), including both targeted (tNGS) and metagenomic (mNGS) approaches, have been employed [4]. The infection typically spreads contiguously through adjacent tissues without lymphatic or hematogenous dissemination, often forming nodular masses or pseudotumors.

The most common clinical form is cervicofacial actinomycosis, accounting for approximately 50% of cases, followed by thoracic (15–20%) and abdominopelvic (20%) forms [1]. Hepatic actinomycosis (HA) is a rare condition, accounting for only about 5% of all actinomycosis cases. It typically occurs secondarily to an intra-abdominal infection. Clinical manifestations are nonspecific and may include fever, abdominal pain, and weight loss. Radiologic and laboratory findings are often inconclusive, making diagnosis challenging and frequently mistaken for malignancy or other inflammatory conditions [2,5].

IgG4-related disease (IgG4-RD) is a multi-organ immune-mediated condition characterized by tissue infiltration of IgG4-positive plasma cells, storiform fibrosis, and often elevated serum IgG4 levels [6]. It can affect a wide range of organs, most commonly the pancreas, biliary tree, salivary glands, kidneys, lungs, and retroperitoneum [7]. Hepatic involvement, though less frequent, usually presents as IgG4-related sclerosing cholangitis, hepatic inflammatory pseudotumor, or, less commonly, IgG4-related hepatitis [8,9]. Hepatic involvement in IgG4-RD, similar to HA, can present as a pseudo-tumoral lesion, frequently resembling malignancy [10,11]. Both IgG4-RD and hepatic actinomycosis are rare entities that may share similar clinical and radiological features, often appearing as mass-like hepatic lesions. This overlap can lead to misdiagnosis; however, accurate differentiation is essential, as the underlying etiologies differ significantly and require markedly different therapeutic approaches—immune modulation for IgG4-RD [7] versus prolonged antibiotic therapy for actinomycosis [2].

## 2. Case Presentation

We report the first known case in the Western world of a rare and diagnostically challenging condition, in which a patient was diagnosed with both primary hepatic actinomycosis—an extremely rare condition, accounting for only 5% of all actinomycosis cases—and IgG4-related disease affecting the liver. This unusual overlap highlights the critical role of histopathological examination in distinguishing between infectious and immune-mediated hepatic lesions. Written informed consent was obtained from the patient included in the study.

A 74-year-old man with a medical history significant for high-grade gastric mucosa-associated lymphoid tissue (MALT) lymphoma (status post-gastrectomy and chemotherapy) was referred to our department after undergoing an abdominal ultrasound (US), which incidentally revealed a 55 × 55 × 40 mm focal liver lesion in segments VIII/V. His past medical history also included atrial fibrillation, prior endoscopic retrograde cholangiopancreatography (ERCP) with pancreatic stenting, and laparoscopic cholecystectomy.

The US findings were inconclusive, prompting further imaging with abdominal Magnetic Resonance Imaging (MRI). The MRI confirmed the lesion, showing irregular margins and non-homogeneous intensity on T1-weighted sequences, with a hypointense core and hyperintense peripheral rim on T2-weighted sequences. A similar lesion was also noted between segment VI and the right kidney (Figure 1a,b).

At the initial evaluation, the patient reported mild asthenia and weight loss. Physical examination revealed no abdominal tenderness or hepatosplenomegaly. Laboratory tests showed mildly elevated γ-glutamyl transferase (GGT, 130 U/L) and alkaline phosphatase (ALP, 146 U/L), along with impaired renal function (serum creatinine 2.06 mg/dL; estimated glomerular filtration rate (eGFR) 31 mL/min). C-reactive protein (CRP) was within the normal range (0.2 mg/dL).

Given the concern for MALT lymphoma recurrence, a liver biopsy was performed. Histopathological analysis revealed a plasma cell and neutrophilic infiltrate of the hepatocytes, with a predominant expression of IgG4 on immunohistochemistry. No evidence of malignancy, mycobacterial, or fungal infection was identified (Figure 1c).

According to histological findings and markedly elevated serum IgG4 levels (1680 mg/dL, normal range < 140 mg/dL), a diagnosis of IgG4-related disease (IgG4-RD) with hepatic pseudotumor was initially assumed. Total serum IgG was also elevated at 3360 mg/dL (normal range: 700–1600 mg/dL), while IgM levels were 128 mg/dL (normal range: 40–230 dL). Autoimmune screening revealed negative antinuclear antibodies (ANA), anti-mitochondrial antibodies (AMA), anti-smooth muscle antibodies (SMA), and liver–kidney microsomal type 1 antibodies (LKM-1). Immunosuppressive therapy with prednisone (60 mg daily) was initiated.

However, despite corticosteroid therapy, cholestasis worsened, and the hepatic lesion continued to enlarge, extending into the retrohepatic tissue. Azathioprine was added and titrated to 75 mg/day. Despite this, follow-up computer tomography (CT) imaging after eight months showed further progression, with the lesion growing to 8 × 6.4 cm and infiltrating the lower lobe of the right lung and the posterior diaphragmatic crus.

After a multidisciplinary review, a bone marrow biopsy was performed to exclude lymphoproliferative disease, with negative results. A positron emission tomography (PET) scan revealed intense, heterogeneous fluorodeoxyglucose (FDG) uptake in the hepatic lesion. Steroid therapy was gradually tapered and discontinued after 12 months. Due to worsening cholestasis, ursodeoxycholic acid (UDCA) was introduced at 300 mg three times daily.

At 16 months, a follow-up CT scan showed a slight increase in the liver mass (currently 7 × 8 cm) with focal intrahepatic biliary dilation. A trans jugular liver biopsy (TJLB) revealed a CD3+ T-lymphocyte infiltrate with few plasma cells and no significant IgG4-positive plasma cell population.

Although cholestasis partially improved, the patient’s weight loss persisted, and the hepatic lesion continued to grow. After 14 months, a third liver biopsy was performed. Histopathological examination revealed a dense inflammatory infiltrate composed of granulocytes and IgG4-producing plasma cells. Notably, clusters of Gram-positive filamentous bacteria surrounded by sulfur granules were identified with periodic acid-Schiff (PAS) and hematoxylin-eosin staining (Figure 1d–f), establishing a final diagnosis of hepatic actinomycosis.

In consideration of this new diagnosis, azathioprine was promptly discontinued, and the patient was started on intravenous ampicillin (3 g every 6 h) for two weeks, followed by oral amoxicillin (1 g three times daily) for six months. Radiological follow-up demonstrated complete resolution of hepatic lesions.

At the three-year follow-up, the patient is in good clinical condition, with no significant events reported. MRI findings showed fibrotic sequelae in the right hepatic lobe, no evidence of focal lesions, and persistent mild dilatation of both intrahepatic and extrahepatic bile ducts. The patient is no longer receiving immunosuppressive therapy and is currently maintained only on ursodeoxycholic acid (UDCA), which was continued due to persistently mild elevation of GGT on follow-up blood tests and as a supportive measure to promote bile flow and prevent cholestatic complications, given the patient’s history of biliary tract involvement.

## 3. Methods

To support the discussion of this case, we conducted a systematic review of the literature using the following databases: PubMed/MEDLINE, The Cochrane Library, EMBASE, Web of Science, and Scopus. The search was restricted to English-language publications. Two separate searches were performed. The first used the keywords “hepatic actinomycosis” and “hepatic actinomycosis and liver mass”. The second search used the terms “hepatic actinomycosis and IgG4”. The first search yielded 180 articles, of which 120 were excluded due to duplication across databases, and 57 were excluded based on title screening. The second search resulted in 7 articles, of which 4 were excluded by title. In total, 3 articles appeared in both searches. Relevant case reports were reviewed in detail and are summarized in Table 1. These cases were analyzed for clinical presentation, diagnostic methods and patient outcomes, with particular attention to cases reporting a co-occurrence of hepatic actinomycosis and IgG4-related disease.

## 4. Discussion

This case underscores the diagnostic complexity of hepatic actinomycosis (HA), a rare and often misdiagnosed condition that can mimic malignancy or immune-mediated inflammatory diseases [15]. Our patient presented with an incidentally detected hepatic lesion, constitutional symptoms (asthenia and weight loss), mild laboratory abnormalities indicating cholestasis and renal impairment, and inconclusive imaging findings. Initial histopathological analysis suggested an IgG4-related hepatic pseudotumor, prompting the initiation of immunosuppressive treatment. However, it was only after three biopsies that the definitive diagnosis of hepatic actinomycosis was established, with the identification of sulfur granules and *Actinomyces* organisms.

Hepatic actinomycosis is rare, accounting for only 5% of actinomycosis cases, and is typically secondary to intra-abdominal infections [5]. Risk factors for HA include male sex, age between 20 and 60 years, diabetes mellitus, prior abdominal surgery, trauma, and immunosuppression, though cases in immunocompetent individuals have been reported [16,17,18,19]. A recent review by Chegini et al. [15] on 64 HA cases between 2000 and 2020 identified a male predominance (64%) and a high frequency of immunocompetent presentations (92%). Most of these cases were reported in Asia (37.5%), followed by Europe and the Americas, with the overall mortality rate of about 1%.

The distinctive feature of our case is the diagnostic progression: the initial presentation suggested IgG4-related disease (IgG4-RD), but despite immunosuppressive treatment, the disease continued to progress, ultimately revealing a simultaneous diagnosis of both hepatic actinomycosis and IgG4-RD.

Our literature review identified a few relevant case reports describing hepatic actinomycosis presenting with radiological or histopathological features mimicking malignancy or IgG4-related disease.

Notably, two of these studies—by Lee et al. [12] and Won Chung [13]—describe surgical resections initially prompted by suspected malignancy, with a final diagnosis of actinomycosis made only after histological evaluation of the resected organ. Chung’s case differs from ours in that there was initial radiological improvement, followed by subsequent worsening, suggesting that actinomycosis developed secondary to steroid-induced immunosuppression. In contrast, our patient showed no clinical or radiological improvement despite corticosteroid and azathioprine treatment, supporting the hypothesis of a concurrent presentation of IgG4-RD and hepatic actinomycosis. A third case, reported by Song [14], diagnosed hepatic actinomycosis via biopsy, revealing an IgG4-related inflammatory pseudotumor with concurrent actinomycotic colonies. However, this study did not provide further details on antibiotic therapy or patient outcomes.

Two other relevant but distinct cases of overlap have been reported in the literature. Shibata et al. [20] described the development of a hepatic abscess within an IgG4-related hepatic pseudotumor following corticosteroid therapy, highlighting the risk of secondary infection under immunosuppression. In contrast, Masuda et al. [21] reported a rare instance of hepatic actinomycosis occurring concurrently with intrahepatic cholangiocarcinoma, illustrating a complex diagnostic overlap between infectious and malignant hepatic lesions.

Diagnosing hepatic actinomycosis remains challenging due to its nonspecific clinical manifestations—such as fever, malaise, night sweats, weight loss, and abdominal pain—and radiological findings that often resemble hepatic abscesses, metastases, or primary liver cancers [2,22]. Histopathology remains the gold standard for diagnosis, although early biopsies may yield inconclusive results. Definitive features include sulfur granules and filamentous, Gram-positive bacteria embedded within neutrophilic infiltrates. However, similar granules can also be seen in conditions such as nocardiosis, botryomycosis, and fungal infections such as aspergillosis [23]; furthermore, the frequent administration of empiric antibiotic therapy before histological evaluation can reduce the sensitivity of microbiological identification.

In recent years, molecular diagnostic tools have become valuable adjuncts for identifying *Actinomyces* species, particularly when conventional methods yield inconclusive results. Techniques such as fluorescence in situ hybridization (FISH) and next-generation sequencing (NGS) allow for the direct detection of bacterial DNA from clinical specimens, bypassing the need for culture [24,25]. Among these, 16S rRNA gene sequencing has proven especially effective in identifying anaerobic or fastidious organisms—even in samples with low bacterial load or in formalin-fixed paraffin-embedded (FFPE) tissues [26]. These approaches can significantly enhance diagnostic accuracy in cases with ambiguous clinical or histopathological findings.

However, only a limited number of papers describe cases in which both histopathological and cultural investigations were negative, and the definitive diagnosis of actinomycosis was obtained exclusively by molecular tests. A particularly striking example is the case reported by Murphy et al. [27], describing a patient with fever and myalgia and a recent diagnosis of colorectal cancer, in whom multiple liver lesions were detected. In this patient, percutaneous biopsy and culture of the specimen were inconclusive for both malignancy and microorganisms. A subsequent surgical resection was therefore performed, and the final diagnosis was obtained only after laparoscopy resection through genomic polymerase chain reaction (PCR) analysis, which identified *Actinomyces* DNA. This case illustrates how molecular testing can establish the diagnosis after culture and histology failure and how the absence of early molecular testing may lead to diagnostic delays and potentially unnecessary surgery. Further emblematic case on the role of molecular test in diagnosing Actinomycosis was reported by Hsu et al. [28], who described an immunocompromised HIV-positive patient presenting with a liver abscess. In this case, *Actinomyces odontolyticus* was identified together with *Candida albicans* by Matrix-Assisted Laser Desorption/Ionization-Time of Flight (MALDI-TOF) mass spectrometry seven days after culture. Despite appropriate antibiotic treatment, the patient died three weeks later, underlining how diagnostic delays can significantly impact prognosis and clinical outcomes.

Taken together, these two exemplifying reports (others are described in Table 2) highlight the importance of integrating molecular techniques into the diagnostic workflow of suspected actinomycosis, particularly in liver disease, where the clinical and radiological presentation frequently mimics malignancy or other infectious diseases. In such scenarios, molecular diagnostics can be instrumental in establishing a definitive diagnosis and preventing overtreatment.

However, their use is limited by high cost, technical complexity, and the need for specialized laboratories. Moreover, results must be interpreted within the clinical, histological, and microbiological context, as molecular methods can detect non-viable organisms and are susceptible to contamination or database-related misidentification [29,30,31]. Nevertheless, in some instances—as in our case—the diagnosis can still be established through conventional histological evaluation alone. In our patient, the presence of pathognomonic features, including sulfur granules and filamentous bacteria observed in the third biopsy, allowed for a confident diagnosis without the need for molecular testing. Notably, molecular investigations were not performed on the initial biopsies, as an infectious etiology was not clinically suspected at the time. Thus, while molecular diagnostic approaches are undoubtedly valuable tools in the workup of actinomycosis, their use should be considered on a case-by-case basis—particularly when characteristic histopathological features are not evident.

The treatment of hepatic actinomycosis typically involves prolonged antibiotic therapy, usually high-dose intravenous penicillin or ampicillin, followed by extended oral therapy for 6–12 months [32]. Given the polymicrobial nature of many abdominopelvic infections, broad-spectrum coverage may be warranted initially [2]. Surgical intervention is generally reserved for cases with diagnostic uncertainty, poor antibiotic response, or the presence of large abscesses or complications. Minimally invasive drainage may be considered in select cases [33].

In contrast, IgG4-related disease is a systemic immune-mediated fibroinflammatory disorder characterized by elevated serum IgG4 levels, storiform fibrosis, obliterative phlebitis, and dense lymphoplasmacytic infiltration enriched in IgG4+ plasma cells [34,35]. The liver is among the affected organs, and hepatic manifestations often mimic malignancy on imaging. Diagnosis requires integration of clinical, serological, radiological, and histological findings [36,37]. Although elevated serum IgG4 levels are suggestive of the disease, they are not specific, and biopsy remains essential in atypical or unclear presentations [34,38].

Corticosteroids are the primary treatment for IgG4-related disease, although steroid-sparing agents such as azathioprine or rituximab are often required for long-term disease control [6]. Newer agents targeting B- and T-cell responses are currently being investigated, but robust evidence from randomized controlled trials is still limited [39].

**Table 2 ijms-26-09420-t002:** Reported cases of hepatic actinomycosis diagnosed using molecular techniques.

Study, yr	Country	Patient Characteristics	Histology	Culture	Molecular Method	Outcome
Murphy et al., 2019 [27]	USA	Fever, myalgia, recent colorectal cancer diagnosis; multiple liver lesions	Negative/non-diagnostic	Negative	Genomic PCR on resected tissue: *Actinomyces* DNA detected	Survived; diagnosis only after laparoscopy resection
Chao et al. 2011 [40]	Taiwan	Immunocompetent; 5.7 cm liver mass suspected abscess	Not diagnostic	Initially misidentified as *Rothia dentocariosa*	Partial 16S rRNA sequencing confirmed *A. odontolyticus*	Survived; improved with antibiotics
Hsu et al. 2021 [28]	Taiwan	HIV-positive, immunocompromised; liver abscess	Not diagnostic	Positive after 7 days	MALDI-TOF confirmed *A. odontolyticus* + Candida albicans	Died despite treatment
Uehara et al. 2010 [41]	Japan	Immunocompetent; liver abscess	Granulomatous inflammation, no sulfur granules	A. israelii suspected	16S rRNA sequencing confirmed diagnosis	Survived after surgery + antibiotics
Llenas-Garcia et al., 2012 [42]	Spain	Immunocompetent; liver abscess	Non-specific inflammation	Negative	16S rRNA sequencing confirmed *Actinomyces* spp.	Survived after prolonged antibiotic therapy
Riegert-Jonhson D et al., 2002 [43]	UK	Adult with pyogenic liver abscess	Not diagnostic	Negative	PCR and sequencing identified *A.israelii*	Survived

The table summarizes clinical and diagnostic features, including histology and culture results, and the molecular methods employed to identify *Actinomyces* spp. Outcome refers to survival and treatment response as reported by the authors.

## 5. Conclusions

Hepatic actinomycosis and IgG4-related inflammatory pseudotumors are rare but important differential diagnoses in patients with liver masses mimicking malignancy. In this case, overlapping clinical and histopathological features contributed to a significant diagnostic delay, exacerbated by immunosuppressive therapy that may have facilitated bacterial proliferation. This case underscores the importance of early tissue diagnosis, repeated biopsy in the setting of clinical progression, and a multidisciplinary approach to optimize diagnostic accuracy and guide appropriate therapy.

## Figures and Tables

**Figure 1 ijms-26-09420-f001:**
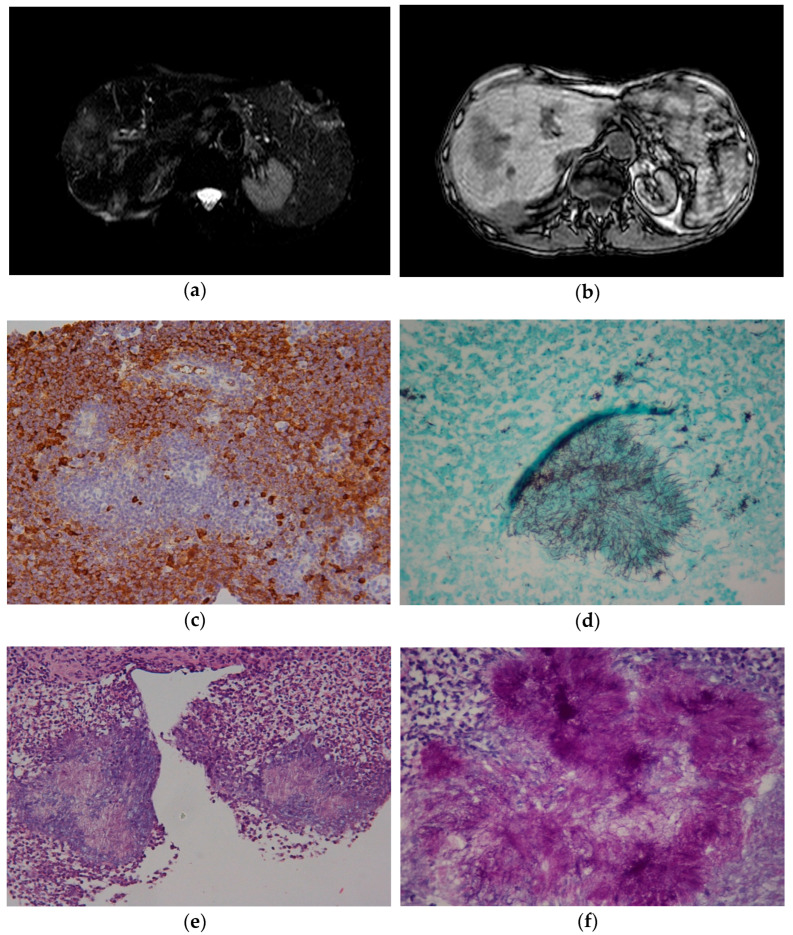
In this figure, the radiological and histological characteristics of actinomycosis are presented. (**a**): MRI Cross-section: lesion with irregular margins and non-homogeneously signal in T1. (Display Field of view (DFOV): 136 × 56.6 cm; native zoom 1×; spatial resolution: ~2.66 mm/pixel). (**b**): MRI cross-section: lesion with a hypointense core and hyperintense edges in T2. (DFOV: 114 × 47.2 cm; native zoom 1×; spatial resolution: ~2.23 mm/pixel). (**c**): IGG4 expression (immunohistochemical analysis, 20×). (**d**): cell aggregates of *Actinomyces* (PAS staining, 40×). (**e**): cell aggregates of *Actinomyces* (standard hematoxylin-eosin staining, 20×). (**f**): cell aggregates of *Actinomyces* (PAS, 40×).

**Table 1 ijms-26-09420-t001:** Clinical and diagnostic characteristics of published cases describing coexisting hepatic actinomycosis and IgG4-related disease, including the present case.

Study, yr	Sex, Age	Clinical Presentation	Risk Factors for Actinomycosis	IgG4-RD Diagnosis	Actinomycosis Diagnosis	Outcome
Lee et al., 2018 [12]	M, 67	Abdominal discomfort, weight loss	Male sex, diabetes mellitus	Histopathology	Histology (surgical specimen)	Positive
Won Chung et al., 2019 [13]	M, 67	Fever, weight loss	Male sex, diabetes mellitus	Serum IgG4, histopathology	Histology (surgical specimen)	Positive
Song et al., 2024 [14]	M, 72	Anorexia, weight loss	Male sex, gastrectomy, prostatectomy	Serum IgG4, histopathology	Histology (liver biopsy)	Not reported
Current study, 2025	M, 74	Asthenia, weight loss	Male sex, gastrectomy, cholecystectomy	Serum IgG4, histopathology	Histology (liver biopsy)	Positive

Outcome refers to clinical evolution, with “Positive” indicating documented improvement or recovery.

## Data Availability

No new data were created or analyzed in this study. Data sharing is not applicable to this article.

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
