# Peer review of "Concomitant Autoimmune Liver Disease and Hepatic Actinomycosis: A Diagnostic Challenge—Brief Report and Review of the Literature"

_ijms, 2025, doi:10.3390/ijms26199420_

Round 1

Reviewer 1 Report

Comments and Suggestions for Authors

In this interesting case report, the authors should

  • exchange "detrimental" with a correct word (it is just a translantion mistake)
  • clarify, if they study protocoll was really approved by the ethic's committee (there does not seem to be a protocoll which has to be approved)
  • discuss, why they proceeded with immunosuppressive therapy such as azathioprine although the patient did not respond to this therapy
  • discuss, if there is a reason why the patient still receives Urso (there does not seem to be one)
  • rephrase the caption to panel 1, which is not a "table"
  • remove the treatment section from table 1, which is misleading, because it does not contain information on the antbiotic therapy
  • remove the information about how the literature research was done from the discussion section to the methods section

Author Response

We would like to thank you for the time and effort dedicated to the review of our manuscript titled Concomitant Autoimmune Liver Disease and Hepatic Actinomycosis: A Diagnostic Challenge – Brief Report and Systematic Review of the Literature (Manuscript ID: ijms-3859701). We sincerely appreciate your thoughtful and constructive comments, which have been very helpful in improving the quality and clarity of our work.

We have carefully considered all your suggestions and have revised the manuscript accordingly. In the following pages, we provide a detailed, point-by-point response to each comment. For clarity, the reviewers’ comments are reported, followed by our responses in regular font. All changes in the revised manuscript have been highlighted in red.

We hope that the revised version of our manuscript meets your expectations, and we remain at your disposal for any further clarification.

Thank you again for your valuable feedback and support.

Reviewer #1 – Comment 1

Exchange "detrimental" with a correct word (it is just a translation mistake)
Author response:

Thank you for pointing this out. We agree that “detrimental” was not the appropriate term in this context. The sentences have been revised accordingly to use more accurate and contextually appropriate words (“not conclusive” page 1, abstract section, line 18 and “inconclusive” pag 18, abstract section, line 23) in the updated manuscript.

Reviewer #1 – Comment 2: 

Clarify if the study protocol was really approved by the ethics committee (there does not seem to be a protocol which has to be approved)
Author response:

We fully agree with the reviewer’s observation. There was no formal study protocol requiring ethics committee approval, as this is a retrospective case report involving a single anonymized patient with informed consent obtained. The initial mention of ethical approval was inserted by mistake. In line with the journal’s guidelines, this statement can be removed from the manuscript (page 8, line 296).

Reviewer #1 – Comment 3: 

Discuss why they proceeded with immunosuppressive therapy such as azathioprine although the patient did not respond to this therapy
Author response:

We thank the Reviewer for this important comment. Azathioprine was introduced as a steroid-sparing agent in the context of a working diagnosis of IgG4-related hepatic pseudotumor, after partial biochemical response to corticosteroids and due to the need for long-term immunosuppression. At the time, histological and serological findings supported this diagnosis, and no signs of infection were present. Despite the lack of radiological improvement, the persistence of systemic symptoms and inflammatory markers led us to maintain immunosuppressive therapy, while continuing diagnostic work-up to rule out alternative causes. Once actinomycosis was diagnosed on the third liver biopsy, azathioprine was immediately discontinued and appropriate antibiotic therapy initiated, leading to complete resolution of the lesion (page 3, case presentation paragraph, line 124).

Reviewer #1 – Comment 4: Discuss if there is a reason why the patient still receives Urso (there does not seem to be one)
Author response:
We thank the Reviewer for this important observation. We agree that, in the absence of active biliary injury, long-term use of ursodeoxycholic acid (UDCA) may appear unnecessary. However, UDCA was continued due to the persistent mild elevation of GGT on follow-up blood tests, along with MRI findings showing mild intra- and extra-hepatic bile duct dilatation. Although the patient remains clinically stable, the use of UDCA was considered appropriate for its potential hepato- and cholangioprotective effects, particularly in the context of prior biliary tract involvement. This point has been clarified in the revised manuscript (page 3, case presentation paragraph, lines 132-135).

Reviewer #1 – Comment 5: 

Rephrase the caption to panel 1, which is not a "table"
Author response:

Thank you for your attention to detail. The caption to Panel 1 has been revised accordingly to reflect that it is an image, not a table (page 4, case presentation paragraph, line 136).

Reviewer #1 – Comment 6: 

Remove the treatment section from Table 1, which is misleading because it does not contain information on the antibiotic therapy
Author response:

We agree with the Reviewer and have removed the “Treatment” column from Table 1 to avoid misleading or incomplete representation of therapeutic regimens.

Reviewer #1 – Comment 7: 

Remove the information about how the literature research was done from the discussion section to the methods section
Author response:

Thank you for this valuable suggestion. We agree that the description of the literature search is more appropriate in the Methods section. Accordingly, we have created a new “Methods” section (page 4, paragraph 3, line 142) in the manuscript, where we describe in detail the databases searched, keywords used, inclusion and exclusion criteria, and the number of articles retrieved and selected. The corresponding content has been removed from the Discussion section to avoid redundancy and maintain structural consistency. We believe this change improves the clarity and organization of the manuscript.

Reviewer #1 Comment - : “Are all figures and tables clear and well-presented? Can be improved”

Author Response: Thank you for this constructive feedback. We agree that enhancing the clarity and readability of our tables is essential for effectively communicating our findings. Accordingly, we have revised both Table 1 and Table 2 to improve their presentation and ensure that critical information is conveyed clearly and consistently.

Reviewer 2 Report

Comments and Suggestions for Authors

This is an excellent case report, especially regarding the autoimmune features. I have only two minor questions.

Minor points:

  1. You describe correctly autoimmunity features. Please provide data of IgG and autoimmune parameters such as ANA and SMA.
  2. What happend with the liver enzymes following treatment by the antibiotic? Dis you see features of DILI under these conditions?

Author Response

Response to Reviewers – Manuscript ID: ijms-3859701

We would like to thank you for the time and effort dedicated to the review of our manuscript titled Concomitant Autoimmune Liver Disease and Hepatic Actinomycosis: A Diagnostic Challenge – Brief Report and Systematic Review of the Literature (Manuscript ID: ijms-3859701). We sincerely appreciate your thoughtful and constructive comments, which have been very helpful in improving the quality and clarity of our work.

We have carefully considered all your suggestions and have revised the manuscript accordingly. In the following pages, we provide a detailed, point-by-point response to each comment. For clarity, the reviewers’ comments are reported, followed by our responses in regular font. All changes in the revised manuscript have been highlighted in red.

We hope that the revised version of our manuscript meets your expectations, and we remain at your disposal for any further clarification.

Thank you again for your valuable feedback and support.

Reviewer #2 – Comment 1

You describe correctly autoimmunity features. Please provide data of IgG and autoimmune parameters such as ANA and SMA.
Author response:
We thank the Reviewer for this valuable comment. As requested, we have added the laboratory data concerning total serum IgG and IgM levels, as well as the results of autoimmune screening (ANA, SMA, AMA, and LKM-1), to the revised manuscript. Specifically, autoimmune screening revealed negative ANA, AMA, SMA, and LKM-1. These data are now reported in the revised version of the case description (pages 3-4, case presentation paragraph, lines 97-101).

Reviewer #2 – Comment 2

What happened with the liver enzymes following treatment by the antibiotic? Did you see features of DILI under these conditions?
Author response:

We thank the Reviewer for raising this relevant point. Liver enzymes were within normal limits throughout and following the antibiotic treatment. No clinical or laboratory features suggestive of drug-induced liver injury (DILI) were observed during or after the antibiotic course.

Round 2

Reviewer 1 Report

Comments and Suggestions for Authors

Well done!